# Otto Engine for the *q*-State Clock Model

**DOI:** 10.3390/e24020268

**Published:** 2022-02-13

**Authors:** Michel Angelo Aguilera, Francisco José Peña, Oscar Andrés Negrete, Patricio Vargas

**Affiliations:** 1Department of Physics, Universidad Técnica Federico Santa María, Avenida España 1680, Valparaíso 2390123, Chile; michel.aguilera@usm.cl (M.A.A.); francisco.penar@usm.cl (F.J.P.); oscar.negrete@usm.cl (O.A.N.); 2Center for the Development of Nanoscience and Nanotechnology, Santiago 8320000, Chile

**Keywords:** *q*-state clock model, entropy, Berezinskii–Kosterlitz–Thouless transition, Otto engine, mean-field approximation

## Abstract

This present work explores the performance of a thermal–magnetic engine of Otto type, considering as a working substance an effective interacting spin model corresponding to the q− state clock model. We obtain all the thermodynamic quantities for the *q* = 2, 4, 6, and 8 cases in a small lattice size (3×3 with free boundary conditions) by using the exact partition function calculated from the energies of all the accessible microstates of the system. The extension to bigger lattices was performed using the mean-field approximation. Our results indicate that the total work extraction of the cycle is highest for the q=4 case, while the performance for the Ising model (q=2) is the lowest of all cases studied. These results are strongly linked with the phase diagram of the working substance and the location of the cycle in the different magnetic phases present, where we find that the transition from a ferromagnetic to a paramagnetic phase extracts more work than one of the Berezinskii–Kosterlitz–Thouless to paramagnetic type. Additionally, as the size of the lattice increases, the extraction work is lower than smaller lattices for all values of *q* presented in this study.

## 1. Introduction

The Otto cycle, widely used by the automotive industry, is today one of the most studied cycles theoretically and experimentally in thermodynamics [1,2,3,4,5,6,7,8,9,10,11,12,13,14,15,16,17,18,19,20,21,22]. This is due to two fundamental reasons: The first is that the efficiency depends on the properties of the working substance, and the second is that its execution stages separate the contributions of work and heat [23,24]. The standard Otto cycle consists of two isochoric trajectories and two isentropic trajectories. In the case where the control parameter is the external magnetic field, the isochoric paths are constant magnetic field processes. In this context, the performance of various working substances operating under an Otto cycle where the control parameter corresponds to an external magnetic field has been studied, where we highlight quantum dots [25], graphene quantum dots [26], multiferroic chain [27,28], twisted bilayer grapehene [29], and two-spin systems with the Dzyaloshinski–Moriya interaction [30], among others.

On the other hand, the *q*-state clock model is the discrete version of the famous 2D XY model [31,32,33,34], which is probably the most extensively studied example showing the Berezinskii–Kosterlitz–Thouless (BKT) transition in the presence of a frustrated quenched disordered phase [35,36,37,38,39,40]. The *q*-state clock model is one of many magnetic models to mimic the thermodynamics of some materials, and it can be viewed as a classical Heisenberg spins model with very strong planar anisotropy [31].

One way to characterize the phase transitions of the *q*-clock state model is through the maxima obtained in the specific heat as a function of temperature. Each location of a maximum of the specific heat on the temperature axis will represent a value for a so-called *critical temperature*. It has been shown [35,36,37,38,39,40] (in the absence of an external magnetic field) that for the *q*-clock state model, values q≥5 (where *q* represents the number of possible orientations that the spins can take), the specific heat presents two maxima. The first maximum corresponds to a transition from a ferromagnetic phase (FP) to a BKT phase, while the second maximum corresponds to a transition from BKT to a paramagnetic disordered phase (PP) [31].

In this research, we propose to study the work and efficiency of an Otto engine whose working substance is an interacting spin system based on the well-known *q*-state clock model. For this purpose, a complete analysis of the thermodynamics of small lattice systems will be made by exact calculations, and the mean-field approximation will be used for large lattice sizes. Phase diagrams will be calculated for a correct analysis to establish the cycle’s operating range and what kind of transitions are involved. In addition, the effects of lattice size on the cycle performance are studied. In particular, for our simulations, it is found that the model with four spin degrees of freedom is the one with the best performance.

This article is organized in the following way: The next section describes the system. Section 3 covers the calculations of thermodynamics. Section 4 explains the model of the engine proposal. Section 5 is devoted to the presentation of the phase diagram of the system. Section 6 is oriented to understand where the Otto engine simulations are positioned in the phase diagram of the proposed working substance. Section 7 presents the results and their discussion, and finally, Section 8 includes the main conclusions of this paper.

## 2. Spin Model

### q-State Clock Model

The working substance under study corresponds to the *q*-states clock model on a two-dimensional (2D) square lattice of dimensions L×L=N, where the local magnetic moment or “spin” $S_{i}$ at site *i* can point in any of *q* directions in a given plane. Then, $S_{i}$ is a 2D vector, i.e., $S_{i}=\left(\cos\left(\frac{2\pi }{q}k\right),\sin\left(\frac{2\pi }{q}k\right)\right)$, where k=0,1,…q−1, with equal probability. The magnitude of $S_{i}$ is chosen to be the unity.

The isotropic Hamiltonian for such a system can be written as [31,32,33,34]:(1)$$H=-\sum_{\langle i,j\rangle }J\left({\vec{S}}_{i}\cdot {\vec{S}}_{j}\right)-\sum_{i}\vec{B}\cdot {\vec{S}}_{i}\;,$$
where J>0 is the ferromagnetic exchange interaction to nearest neighbors; the sum runs over all pairs of nearest neighbors (i,j), which is indicated by the symbol 〈i,j〉 under the summation symbol. $\vec{B}$ is an external field applied along one direction in the plane. In this work, we used arbitrary units choosing J=1, making all the calculated quantities be in terms of the exchange energy constant. Figure 1 presents an example for a 3×3 lattice of this model.

For the thermodynamic analysis of this model, we will perform two types of calculations to derive the partition function of the system. The first is an exact calculation of all the accessible microstates of the system for a 3×3 lattice for q=2,4,6, and 8. For the same values of *q* studied, for lattices of size up to 256×256, mean-field theory will be employed. Both calculations will be detailed below.

## 3. Thermal Averages: Thermodynamics

### 3.1. Microstates

Let us start the thermodynamic discussion for the 3×3 finite lattice with free boundary conditions. We will use an approximation that we will call *exact* because it corresponds to an exact diagonalization of the Hamiltonian given by Equation (Equation 1) and the corresponding calculation of all the possible microstates that the system possesses.

The partition function is obtained as follows
(2)$$Z\left(T,B\right)=\sum_{n=1}^{\lambda }C_{n}e^{-\frac{E_{n}}{T}},$$
where the coefficients $C_{n}$ correspond to all possible spin configurations compatible with an energy $E_{n}$ coming from the Hamiltoninan of Equation (Equation 1) (basically representing the degeneracy of the each energy level of the system) and λ is the number of different values of energy levels. For all our calculations, we will use the Boltzmann constant $k_{B}=1$, which means that temperature and energy are in the same units.

The number of microstates depends on the freedom of spin orientations, the size of the lattice, and the external magnetic field, in other words, on *q*, *L*, and B. Each spin has *q* potential states, and therefore, all the possible self-energies of the system are given by
(3)$$N_{states}=q^{L\times L}.$$

An example of the number of microstates for finite lattice systems is presented in Table 1. An example of the possible spin configurations for q=8 in a 3×3 and 16×16 lattice size is shown in Figure 2. In Table 1, we note that for L=3, the Ising model must consider 512 lattice configurations to estimate the partition function. It is logical to think, seeing the numbers presented in this table, that for large lattice sizes, the computational cost of these calculations is not viable at present, and therefore, alternative methods such as Monte Carlo simulations and the mean-field approximation are used. We will use the latter for a larger lattice than the 3×3 size.

### 3.2. Mean Field Approximation

The mean-field theory is based on the assumption that the fluctuations around the average value of the order parameter (in this case, the magnetization $\vec{m}$ ) are so small that they can be neglected. The first term of the Hamiltonian of Equation (Equation 1) that corresponds to the interaction term between the spin of the lattice in different sites is modified by performing the following approximations.

We can write the spin term as follows
(4)$${\vec{S}}_{j}=\vec{m}+\delta {\vec{S}}_{j},$$
where $\vec{m}$ is the average thermodynamic spin, the same for all sites in the lattice. Therefore, we have
(5)$$\delta {\vec{S}}_{j}={\vec{S}}_{j}-{\vec{m}}_{j}.$$

Thus, the spin–spin interaction term can be written as
(6)$${\vec{S}}_{i}\cdot {\vec{S}}_{j}=-m^{2}+\vec{m}\cdot \left({\vec{S}}_{i}+{\vec{S}}_{j}\right),$$
where we have neglected the square terms of the fluctuation ($O{\left(\delta \vec{S}\right)}^{2}$). Therefore, the interaction term of the Hamiltonian of Equation (Equation 1) (that we call $H_{J}$) can take the form
(7)$$H_{J}=-J\sum_{\langle i,j\rangle }\left(-m^{2}+\vec{m}\cdot \left({\vec{S}}_{i}+{\vec{S}}_{j}\right)\right)=\sum_{i}\left(\frac{z}{2}Jm^{2}-Jz{\vec{m}}_{i}\cdot {\vec{S}}_{i}\right),$$
where now, the sum runs for each site in the lattice and *z* are the effective nearest neighbors of the model (see Figure 3 for an example). Consequently, we can define a Hamiltonian per site given by the structure
(8)$$h_{i}=\frac{z}{2}Jm^{2}-Jz{\vec{m}}_{i}\cdot {\vec{S}}_{i}-\vec{B}\cdot {\vec{S}}_{i}.$$

Now, we can calculate the partition function per site, which will depend on *q*, B, *m*, and *T* given by
(9)$$Z\left(q,B,m,T\right)=\sum_{q}e^{-\frac{\xi (q,B,m)}{T}},$$
where ξ(q,B,m) is the energy per site coming from the Hamiltonian of Equation (Equation 8).

We found that for mean field with a number of nearest neighbors z=4, in the framework of a small system with lattice L×L=3×3, that the internal energy behaves differently from the one obtained in an exact approximation. It is proposed to find a number of effective nearest neighbors that fits the approximation through an optimization. By releasing the number of neighbors, $z_{eff}\in \;{ℜ}^{+}$, and minimizing the internal energy difference (between the exact and approximate case via mean field), it was found that for L=3, the optimal number of neighbors was $z_{eff}=2.67$ (for all values of *q*). This can be seen in Figure 4 for the Ising model (q=2, as an example) on a 3×3 lattice where the internal energy is shown for an external field B=1 and B=4 for different values of *z* from z=0.1 to z=4.

Amplifying the above qualitatively, we propose an expression for the number of near neighbors effective that adjusts according to the weighting of the effect of non-interacting edges in the system when the lattice has a generic resolution L×L. For the square lattice with up, down, left, and right near neighbors, the effective neighbors for a central spin for the mean field is determined by the following expression (independent of *q*).
(10)$$z_{eff}=\frac{4\times {\left(L-2\right)}^{2}+3\times 4\times \left(L-2\right)+2\times 4}{L^{2}}.$$

### 3.3. Thermodynamic Relations

Once the partition function of the system has been obtained by either of the two approaches discussed above, it is possible to compute all the thermodynamical observables in a general way through the expressions (with $k_{B}=1$)
(11)F=−TlnZ,
(12)$$U=T^{2}\frac{\partial \ln Z}{\partial T},$$
and
(13)$$C=\frac{\partial U}{\partial T},$$
where *F* is the Helmholtz free energy, *U* is the internal energy, and C is the specific heat at constant magnetic field. In addition, with the differential expression of Helmholtz free energy given by dF=−SdT−MdB, we can obtain the entropy and the magnetization of the system given by
(14)$$S=-\frac{\partial F}{\partial T};\;M=-\frac{\partial F}{\partial B}.$$

## 4. Otto Engine

### Description of Otto Engine

The standard Otto engine is a quasi-static cycle (which means there is always thermodynamic equilibrium) that considers two isochoric and two adiabatic processes [14,15,16,17,25,26]. In our case, the isochoric stages are replaced by constant magnetic field processes. Therefore, the entropy versus external magnetic field diagram is represented by a rectangle, as shown in Figure 5. The Otto cycle processes are detailed below.

(1) Adiabatic compression (stage A→B). The system, which is initially at a temperature $T_{l}$ and an external field $B_{1}$, is subjected to an increase in the external magnetic field up to a value $B_{2}$ without exchanging heat with its surroundings. From the first law of thermodynamics, we will then have that the total work done in the process is given by:(15)$$W_{A\to B}=U_{B}\left(T_{B},B_{2}\right)-U_{A}\left(T_{l},B_{1}\right),$$
where *U* corresponds to the internal energy of the system given by Equation (Equation 12). When the external magnetic field changes from $B_{1}$ to $B_{2}$, the evolution of the temperature in the adiabatic process is not free and must be governed by the condition of entropy equality given by
(16)$$S\left(T_{l},B_{1}\right)=S\left(T_{B},B_{2}\right),$$
where *S* it is defined by Equation (Equation 14).

(2) Isochoric heating stroke (stage B→C). The system is placed in contact with a thermal reservoir at temperature $T=T_{h}$ until the working substance reaches thermal equilibrium with the reservoir. This process is carried out at a constant magnetic field, and there is no work done during its execution. There is only heat exchange between the working substance and the reservoir given by
(17)$$Q_{in}=U_{C}\left(T_{h},B_{2}\right)-U_{B}\left(T_{B},B_{2}\right).$$

(3) Adiabatic expansion (stage C→D). The system is disconnected from the thermal reservoir and subjected to a change in the external magnetic field from $B_{2}$ to $B_{1}$ without exchanging heat with its surroundings. There is only work done at this stage given by the expression
(18)$$W_{C\to D}=U_{D}\left(T_{D},B_{1}\right)-U_{C}\left(T_{h},B_{2}\right).$$

Again, the temperature at this stage does not evolve freely and depends on the constant entropy condition in this case given by
(19)$$S\left(T_{h},B_{2}\right)=S\left(T_{D},B_{1}\right).$$

(4) Isochoric cooling stroke (stage D→A). Finally, the system is put in contact with a thermal reservoir at temperature $T=T_{l}$ until thermal equilibrium with the reservoir is reached. The process is performed at constant magnetic field $B=B_{1}$ and there is no work done during this stage: only heat exchange between the working substance and the reservoir. Then, the heat output is defined as
(20)$$Q_{out}=U_{A}\left(T_{l},B_{1}\right)-U_{D}\left(T_{D},B_{1}\right).$$

The efficiency of a thermodynamic engine is defined by
(21)$$\eta =\frac{|W_{total}|}{Q_{in}}.$$

In our case, $W_{total}$ is given by
(22)$$W_{total}=W_{A\to B}+W_{C\to D},$$
where $W_{A\to B}$ and $W_{C\to D}$ are given by Equation (Equation 15) and Equation (Equation 18), respectively.

Simulations of the standard Otto engine are obtained by fixing the values of $T_{l}$, $T_{h}$, and $B_{l}$ and infinitesimally moving the $B_{h}$ field to an arbitrary physically possible value. That is why the points of the A and C states in the cycle are well-determined values in the calculations. As mentioned above, the *q*-state clock model has one phase transition for q≤4 and two-phase transitions for q≥5. This is why it is essential to know where points A and C are located in our simulations, as this will indicate whether we are operating an engine through these phase transitions. In the following section, we will calculate the so-called *phase diagrams* to select and interpret correctly the region where our motor operates for the different values of *q* that will be presented in the results.

## 5. Phase Diagram

The maximum values of the heat capacity define phases of magnetic order. Therefore, a qualitative analysis of the behavior of the specific heat concerning temperature and the external magnetic field is proposed. We can see an example of the maxima in specific heat for a 3×3 lattice for the exact evaluation in the cases of q=2 and q=4 in Figure 6a,b and for q=6 and q=8 in Figure 6c,d, respectively. In these figures, it can be clearly seen that for q≥5, the specific heat has two maxima, which is indicative of a double phase transition. However, this two peaks observed for q=6 and q=8 (Figure 6c,d, respectively) shift to higher temperatures as the external magnetic field increases. This behavior is easily explained by the external field favoring ordered phases (FM and BKT) over disordered ones, and therefore, the transition temperatures increase with the strength of the external field [31]. These results of thermodynamics observable are consistent with those reported in previous work in the literature [31,32,33,34].

Obtaining the curve representing the boundary between phases is based on maximizing the heat capacity for a given field and saving the pair of points $\left(B_{fix},T_{cr}\right)$ for each model. In Figure 7a,b, we visualize the phase diagram for q=2 and q=4, while for Figure 7c,d, we show the phase diagram for q=6 and q=8, respectively. Both figures show calculations with exact approximation for a small 3×3 lattice. Figure 7a,b represent the specific heat maxima presented in Figure 6a,b showing its FP and PP phases as expected for these values of *q*, whereas Figure 7c,d visualize the BKT phase for q=6 and q=8 in accordance with the specific heat figures shown in Figure 6c,d.

In Figure 7a, for the Ising model (q=2), we notice that under the dotted curve, we are in an ordered FP region (blue zone). As the temperature increases, the spins start to disorder until reaching PP (red zone). For q=4 (Figure 7b), we notice a similar transition, with the difference of needing lower temperatures to achieve disorder. The phase diagram for q=6 and q=8 presented in Figure 7c and Figure 7d, respectively, shows three clear phases for q=8, while for q=6, all three phases are present only up to an external magnetic field close to B=1.5. For higher magnetic fields, in the case of q=6, only a transition from FP to PP is present. This last characteristic is in coherence with the specific heat plots shown in Figure 6c where we see that for fields higher than B=1, in this case, B=2, B=3, and B=4 shown in that figure (lemon-green, yellow, and orange line, respectively), a peak in specific heat is lost compared to that shown for B=0 (blue line) and B=1 (purple line). Consequently, we only have a transition from FP to PP type as the temperature increases for this case studied.

## 6. Cycle Reservoirs

Having two phase transitions for q≥5 in the model, positioning the reservoirs deserves a little analysis in favor of understanding how many transitions we will deal with throughout the cycle. For this, it is useful to unify Figure 7a–d and plot the location of the cold and hot reservoirs as a point and a horizontal line on that figure, respectively. This is presented in Figure 8, where we observe that the selection of the cold (point A of the cycle) and hot (point C of the cycle) reservoir for our simulations is given by the points
(23)$$\mathrm{Point}\;A\equiv \left(B_{1}=1.0,T_{l}=0.6\right)$$
(24)$$\mathrm{Point}\;C\equiv \left(B_{2}=1.1-4.0,T_{h}=6\right).$$

The selection of these points is based on satisfying three criteria:

(i) The cold reservoir must have an entropy whose value is distinguishable to the accuracy of numerical calculations in order to solve the first adiabatic condition given by Equation (Equation 16).

(ii) At least one phase transition must be included in the cycle.

(iii) Although the study is initiated with the intention that all models, faced with the same hot reservoir, transit between FP and PP, the FP region of q=8 corresponds to a zone with low entropy, which would generate problems associated with the first point of the criteria under discussion. Consequently, we place the cold reservoir in a BKT phase for this case. As we have discussed above, q=6 may present a double phase transition for magnetic field values B<1.5, so we select a cold reservoir that considers dominant only the region of a single maximum in the specific heat for that value of *q*. In this case, it is a zone where only one kind of transition of type BKT to PP exists, which occurs for B<1.5. This is done to have two study cases with FP to PP transitions and two with BKT to PP transitions. In summary, in the results shown in the following section, the proposed magnetic Otto engine for q=2 and q=4 will transit between phases FP and PP, while for q=6 and q=8, it will transit between phases BKT and PP.

Finally, it is essential to mention that as the lattice size increases, our results indicate that the critical temperatures increase for all *q* values studied, which implies that the FP was becoming more prominent. However, with the points selected for the cycle operation, we conserve the types of transitions for each value of *q* that can occur in the engine’s execution.

## 7. Results and Discussion

We will first analyze the behavior of the work and motor efficiency (given by Equation (Equation 22) and Equation (Equation 21) respectively) in a 3×3 lattice with the exact and mean-field approximation for q=2,4,6, and 8. This analysis is presented in Figure 9, wherein panel (a), the total work is presented and (b) shows the system efficiency. Both plots are shown as a function of the variable magnetic field in the system corresponding to $B_{2}$ from value 1.1 to 4. For the total work extraction presented in Figure 9a, we note that the q=2 curve (Ising model, blue-colored curves) has the worst performance. The q=4,6, and 8 curves decrease the total work extraction obtained as *q* increases, with the q=4 curve (lemon-green-colored curves) having the highest work. It is important to note that we noticed a similar result between the exact (solid lines) and mean-field methods (dashed lines), which indicates the consistency of the presented calculations. These differences between the approximations to obtain the thermodynamics of the system decrease as *q* grows. For q=2 and q=4, the exact method performs better than the mean-field results. The above mentioned is reversed for q=6 and q=8, obtaining higher total work than the mean-field approximation. In the case of efficiency, Figure 9b shows that the q=2 case still presents the worst performance of the cases analyzed. It is also the one that present the largest difference between the exact and mean-field calculations. From Figure 9b, we observe that the efficiency for q=8 is the highest of all cases, which is followed by that for q=4, then q=6, and finally q=2. It is important to note that the differences between the efficiencies of the q=4,6, and 8 cases are relatively small. Consequently, if we think of the best performance of the machine that can be intuited from W×η, this will correspond to the q=4 case.

The behavior of the efficiency for the exact case in the 3×3 lattice can be understood if we analyze the difference between the heat input ($Q_{in}$ given by Equation (Equation 17)) and the heat output ($Q_{out}$ given by Equation (Equation 20)) divided by $Q_{in}$ due to the fact that the efficiency can be written as
(25)$$\eta =\frac{Q_{in}-Q_{out}}{Q_{in}}=1-\frac{Q_{out}}{Q_{in}}.$$

In Figure 10, it can be seen that the ratio between $Q_{out}$ and $Q_{in}$ is not as significant for q=2 as it is for the other values of *q* studied, with the largest differences between $Q_{in}$ and $Q_{out}$ being q=4 and q=8. Consequently, a lower efficiency is expected for the Ising model (q=2) with the parameters selected in the study.

In order to see the effects of lattice size on total work and efficiency, we propose to study with the mean-field approximation the case of a lattice of size 256×256 (for all values of q), where the number of effective neighbors, $z_{eff}$ is already close to the value four and the approximation is more robust. These results are shown in Figure 11a,b, wherein (a) we show the total work and (b) the efficiency of the system in comparison in a 3×3 lattice. For the results to be comparable in Figure 11a, we must speak of work per spin, i.e., divide the total work obtained by the number of spins in the lattice. The first result we can appreciate in work per spin is that the larger the lattice, the smaller the amount of extraction work obtained from the cycle for any value of *q*. The q=2 case is still the lowest total work and the most significant difference between the lattice sizes studied. In addition, q=4 continues to show the highest work extraction. In addition, for a large 256×256 lattice, we observe that there are no significant differences at low magnetic fields (up to about $B_{2}=1.7$ ) in the efficiency of the q=4,6, and 8 cases.

We can establish a quantitative relationship between the results obtained for the total work and the location of the operating zone of the cycle by looking at the phase diagrams in Figure 7. If we first analyze the FP–PP-type transitions, corresponding to the q=2 and q=4 cases (panels (a) and (b) of Figure 7), we observe that the ferromagnetic phase involved in the cycle will be larger than that of the q=4 case, where the preponderant phase will be the PP. Comparing the work of q=2 and q=4, it is already known that q=4 presents a higher work than the case of q=2. If we focus on the BKT–PP-type transitions from Figure 7, we notice that the BKT zone of q=6 (panel (c) of Figure 7) involved in the cycle will be smaller than that of q=8 (panel (d) of Figure 7). If we now compare only the work of q=6 and q=8, the case of q=6 extracts more work than q=8. This simple comparison of results is an indication that the total work will be more significant (with the same operating parameters) when we have a smaller portion of the cycle in a sorted zone. In addition, our results indicate that a FP–PP-type transition is more beneficial than a BKT–PP-type transition for the performance of the proposed magnetic motor when a small portion of the cycle is positioned in an FP zone.

The result of why q=4 in the selected parameter region is the best performing can be analyzed through the *W* in each of the adiabatic stages of the cycle. We notice in Figure 12a, which presents the work $W_{A\to B}$ as a function of $B_{2}$, that the differences between each of the curves for the different values of *q* are minor being the smallest the case of q=2 (blue line) and q=4 (lemon-green line). This is mainly because, according to the phase diagram in Figure 7a,b, the process takes place almost entirely in an FP. While in the case of q=6 and q=8, there is a mixture between the FP and BKT phases. However, when we analyze the work $W_{C\to D}$ presented in panel (b) of Figure 12, we realize that almost all work values for different values of *q* are the same except for q=2, which shows (in absolute value) a more significant difference concerning the previous ones. The C→D process is always carried out in a paramagnetic zone for all *q*, as in the case q=2, where point D of the cycle is closer to the transition between the phases. In energetic terms, for the *q*-clock model, being located close to the transition represents higher internal energy (in absolute value). Therefore, the energy difference will be more considerable between points C and D. As the work at this stage is negative, if $W_{C\to D}$ is large, the smaller the total work of the cycle will be.

Finally, we would like to mention that the effect of the *J* parameter on the cycle is not trivial. *J* plays a transcendental role in the critical temperatures and, therefore, in the phase diagram. In the extreme case of J=0 (a free spin system), the work substance for cases with spin degrees of freedom q=4,6, and 8 reduces the total work, while for q=2, the opposite is true (see Figure 13). This is because in the region of operation we selected for our study, it is more difficult (energetically) to change the spin orientation for an Ising-type model than for models with more degrees of freedom. However, when we increase the value of *q* in the selected region of temperature and field, the exchange favors the visitation of the different states of the *q*-clock model, thus reducing the amount of work required to perform the adiabatic process from C to D. Additionally, we note that for the case J=0, q=6 and q=8 show the same extraction work. This is since for larger values of *q*, the difference between the internal energies (only for J=0) are smaller and smaller; therefore, a convergence in the calculation of the total work in the cycle is expected.

## 8. Conclusions

In this work, we have addressed the possibility of operating an Otto engine whose working substance is an interacting spin system corresponding to the *q*-state clock model. For small lattice systems, we have calculated and analyzed the thermodynamics of the system exactly by obtaining all the accessible microstates of the system, while for larger lattices, we have performed the calculations through the mean-field approximation. The working substance used presents one or two phase transitions depending on the degree of freedom of the spin; therefore, the selection of the operating range of the motor cannot be arbitrarily selected, and in our study, we have placed it for the Ising model (q=2) and q=4 from an ordered phase (ferromagnetic phase) to a disordered phase (paramagnetic phase), while for q=6 and q=8, it is from a vortex phase (BKT phase) to a disordered phase. The results for small-size lattices indicate that for the selected operating range, q=4 presents the best performance based on the extraction work and efficiency that can be obtained in the cycle, while the Ising model is the worst performer of all the cases analyzed. When the lattice size is increased, both the efficiency and spin work decrease, but the q=4 case is still the best-performing case. These reported results can be interpreted from the phase diagram of the working substance, which indicates that a smaller portion of the cycle in a ferromagnetic phase would allow a better total work output.

Our result of the optimal work and efficiency for the system with q=4 does not have a general reason. As the energies grow with temperature, it is convenient that the work window (T,B) used in the Otto cycle includes a low entropy ordered zone, i.e., ferromagnetic phase and high entropy in a high-temperature, paramagnetic disordered phase, so that the energy differences make $Q_{in}-Q_{out}$ to be maximal. In our case, this occurs for q=4, but if the (T,B) window of the Otto cycle is varied, we can find that the work is maximized for another value of *q*.

This work is currently undergoing an extension in the context of an *endoreversible* scenario in order to obtain the finite power output of the proposal machine [41] considering, in addition, the anisotropy and dipolar interaction terms, both of which are fundamental in the accurate description of authentic materials.

## Figures and Tables

**Figure 1 entropy-24-00268-f001:**
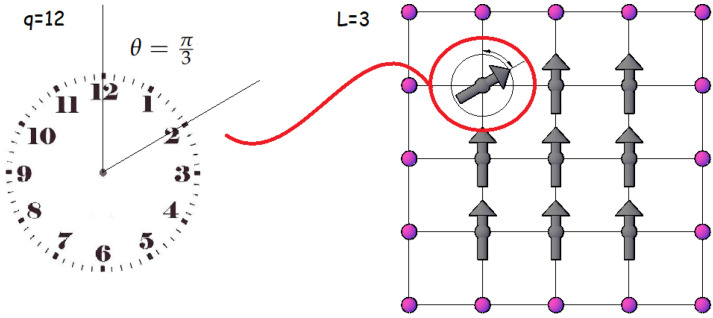
Example of the *q*-state clock model for a 3×3 lattice where the direction of a spin in the lattice is displayed at an angle of $\theta =\frac{\pi }{3}$. The purple circles represent the free boundary conditions in the model.

**Figure 2 entropy-24-00268-f002:**
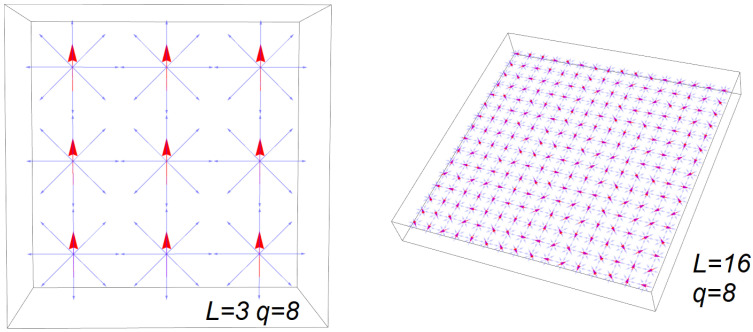
Diagram of spin lattice with q=8 for different lattice size (left L=3 ordered, right L=16 disordered).

**Figure 3 entropy-24-00268-f003:**
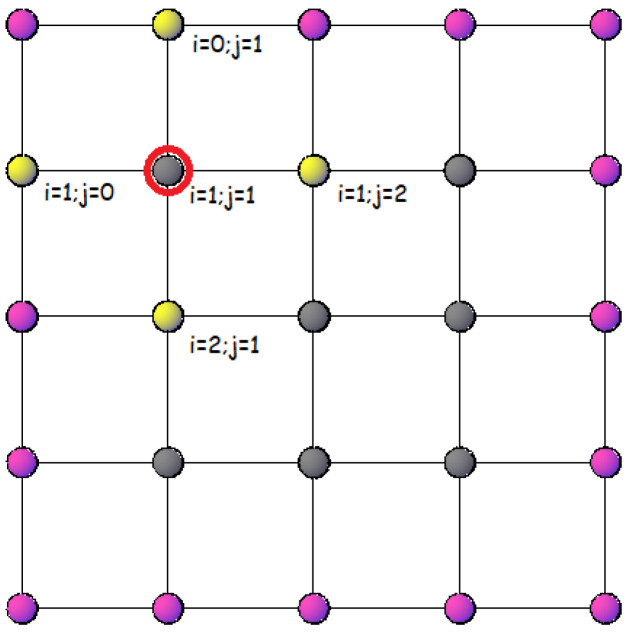
Example of near neighbors (demarcated in yellow) for a spin (1,1) (marked with a circle of red color) in a 3×3 lattice with free boundary conditions.

**Figure 4 entropy-24-00268-f004:**
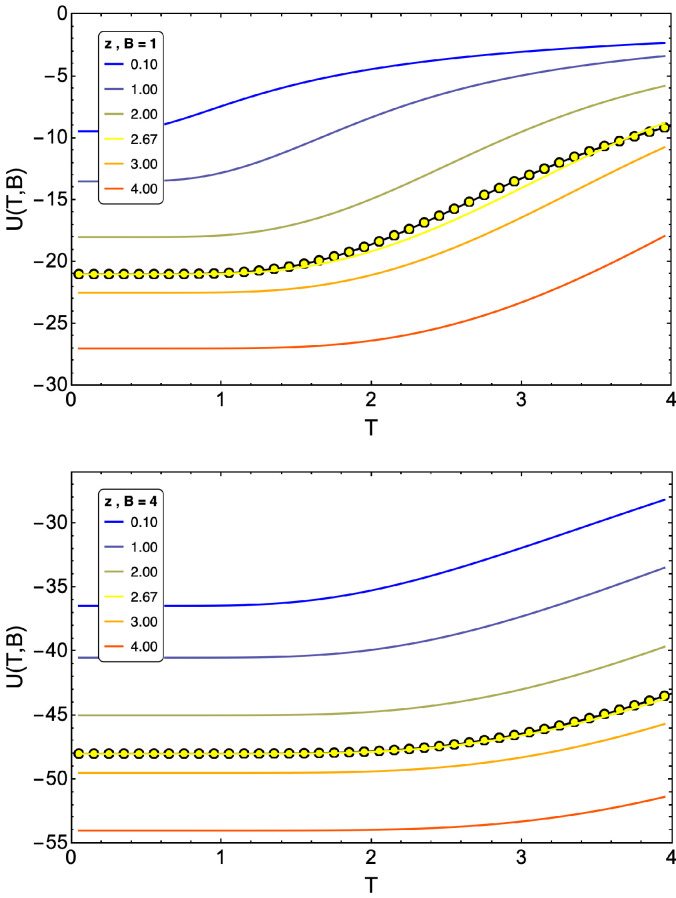
Plots of internal energy for Ising model (q=2), computed exactly (yellow-dotted line) and approximately by mean-field theory distinguishing number of nearest neighbors. We note that z=2.67 fits best when B≥1.

**Figure 5 entropy-24-00268-f005:**
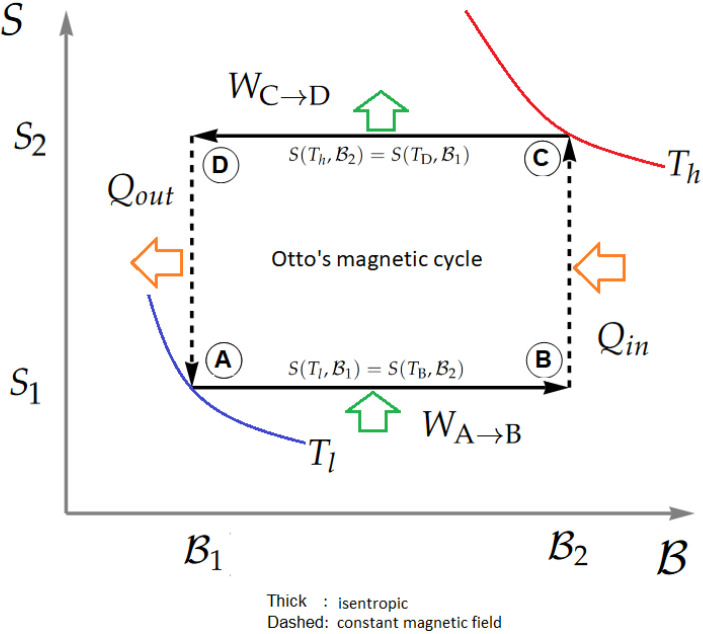
Pictorial representation of the Otto cycle.

**Figure 6 entropy-24-00268-f006:**
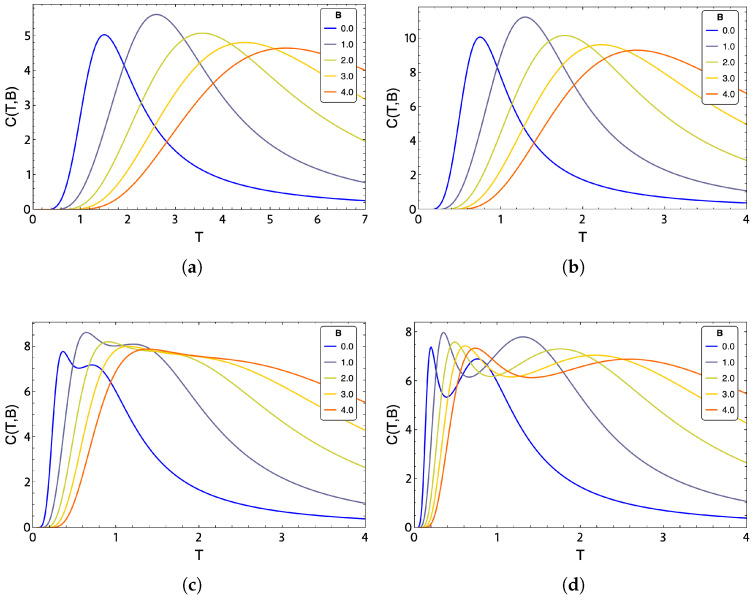
Specific heat as a function of temperature for different values of external magnetic field of values B=0 (blue), B=1 (purple), B=2 (lemon-green), and B=4 (orange) for a 3×3 lattice with different values of *q* parameter. (**a**) Ising model, q=2, (**b**) q=4, (**c**) q=6, and (**d**) q=8.

**Figure 7 entropy-24-00268-f007:**
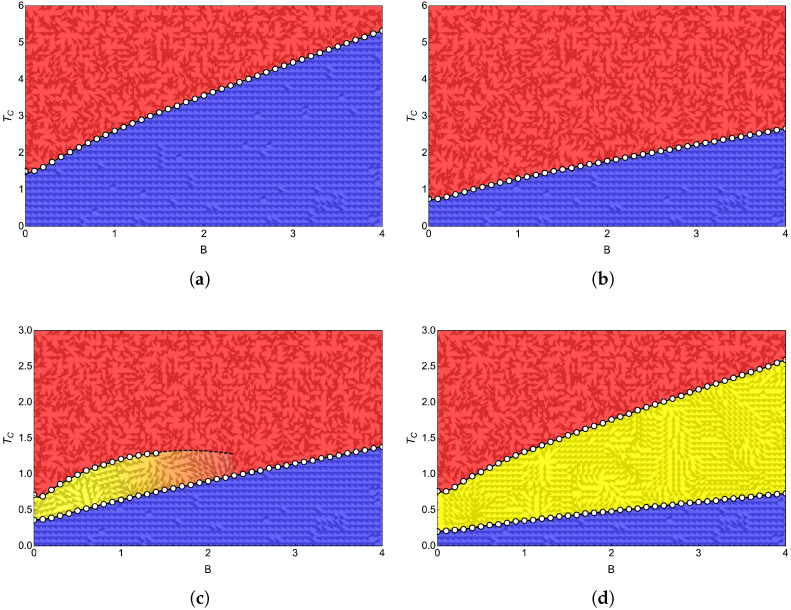
Phase diagrams for (**a**) q=2; (**b**) q=4; (**c**) q=6 and for (**d**) q=8.

**Figure 8 entropy-24-00268-f008:**
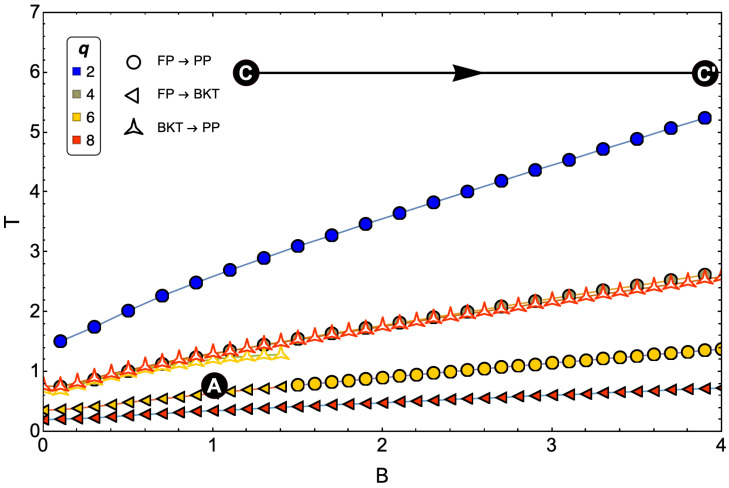
Otto cycle reservoirs selected for the study. The point A with $T_{l}=0.6$ and $B_{1}=1$ shows that q=2 (blue), 4 (lemon-green), and 6 (yellow) is in FP, and for q=8 (red), the cycle starts in BKT phase. Point C is represented with a red line with $T_{h}=6$ given its moving character taking values of $B_{h}$ between 1.1 and 4.0 in 0.1 intervals.

**Figure 9 entropy-24-00268-f009:**
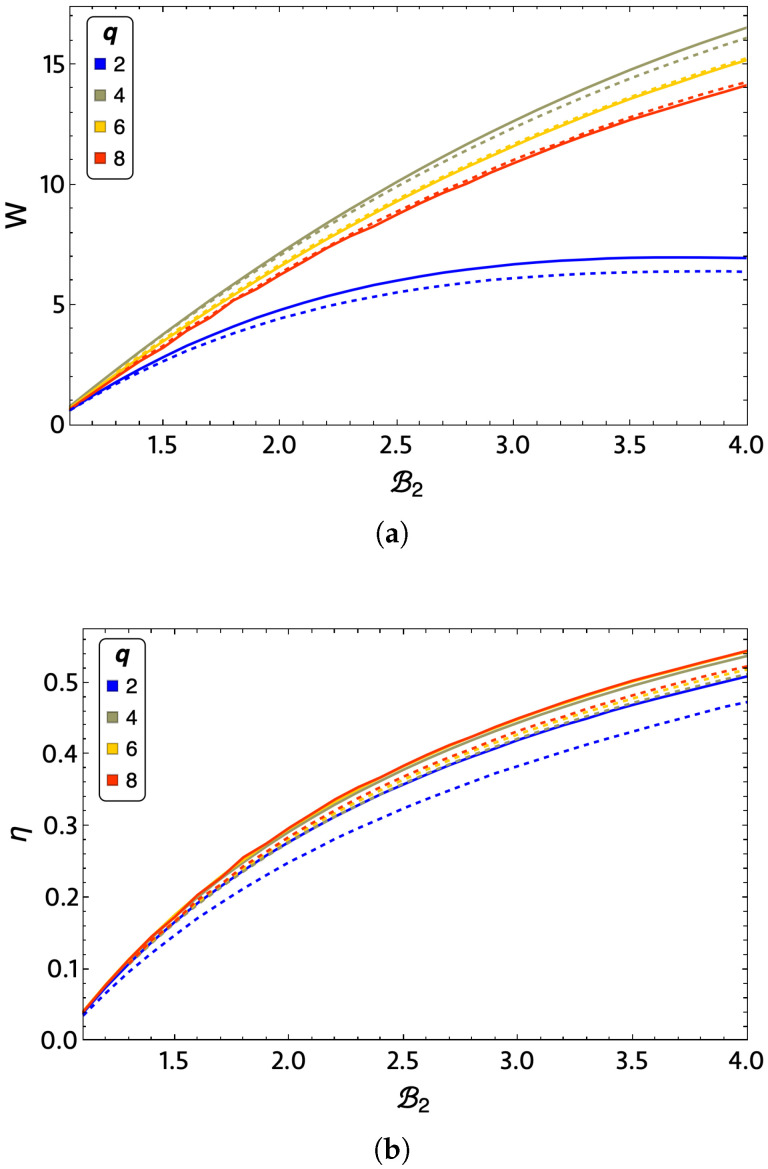
(**a**) Total work and efficiency (**b**) for a 3×3 lattice for q=2 (blue), q=4 (lemon-green), q=6 (yellow), and q=8 (red) for exact calculations (solid line) and mean-field approximation (dashed line) as a function of external magnetic field $B_{2}$.

**Figure 10 entropy-24-00268-f010:**
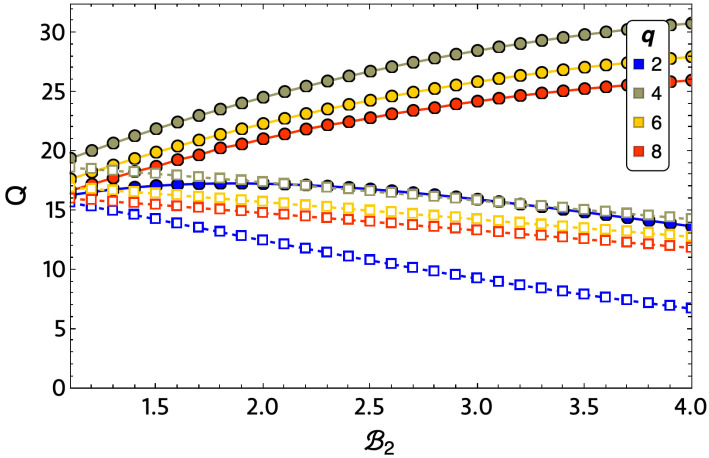
Heat input ($Q_{in}$, dotted line) and heat output ($Q_{out}$, dashed line) for a small lattice of 3×3 for exact calculations of q=2 (blue lines), 4 (lemon-green lines), 6 (yellow lines), and 8 (red lines).

**Figure 11 entropy-24-00268-f011:**
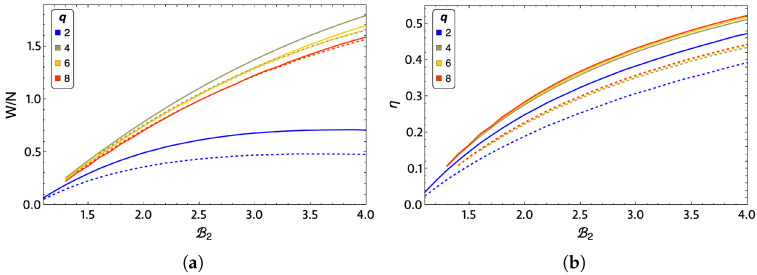
(**a**) Work per spin and efficiency (**b**) for a 3×3 lattice and a 256×256 lattice for different values of *q*: 2 (blue), 4 (lemon-green), 6 (yellow), and 8 (red).

**Figure 12 entropy-24-00268-f012:**
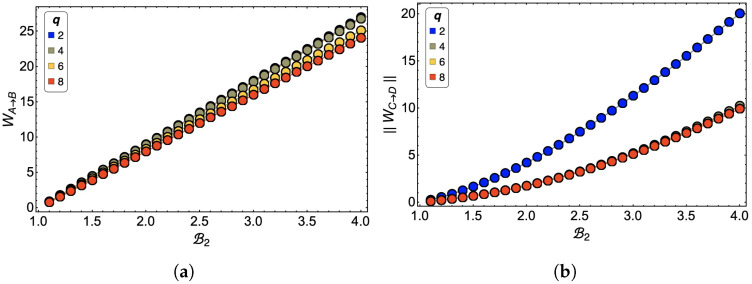
(**a**) Work in the first adiabatic trajectory $W_{A\to B}$ and (**b**) in the second adiabatic trajectory $W_{C\to D}$ for a 3×3 lattice size.

**Figure 13 entropy-24-00268-f013:**
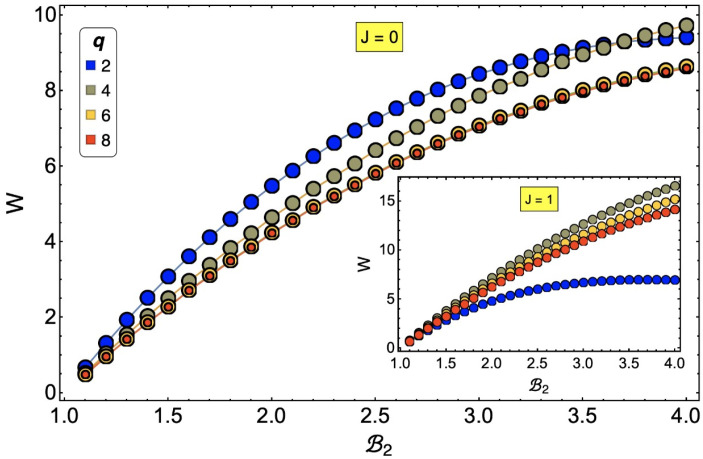
Total work for a 3×3 lattice as a function of external magnetic field $B_{2}$ for different values of *q* for J=0 and J=1 (inset).

**Table 1 entropy-24-00268-t001:** Table with number of microstates according to *q* and *L*.

*L*	*q*	$N_{\mathrm{states}}$	*L*	*q*	$N_{\mathrm{states}}$
3	2	512	8	2	1.84467 $\times \;10^{19}$
3	4	262,144	8	4	3.40282 $\times \;10^{38}$
3	6	10,077,696	8	6	6.33403 $\times \;10^{49}$
3	8	134,217,728	8	8	6.2771 $\times \;10^{57}$
4	2	65,536	16	2	1.15792 $\times \;10^{77}$
4	4	4,294,967,296	16	4	1.3408 $\times \;10^{154}$
4	6	2.82111 $\times \;10^{12}$	16	6	1.6096 $\times \;10^{199}$
4	8	2.81475 $\times \;10^{14}$	16	8	1.5525 $\times \;10^{231}$

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
