# Peer review of "Otto Engine for the q-State Clock Model"

_entropy, 2022, doi:10.3390/e24020268_

Round 1
Reviewer 1 Report
The paper describes an Otto engine cycle across a phase transition. The model analysed is based on the q-state clock model which allows to interpolate from the 2-D Ising model to more complex interactions.
The results are interesting. Thinking about the cycle; optimal work could be achieved if the work required fro the cold compression could be decreased by choosing a phase which minimizes the cost of compression. The opposite should work for the hot expansion where the phase should be chosen to maximize the output work.
Author Response
Dear Referee, please see attached pdf,
best regards,
Francisco Peña on behalf of all the authors.

Reviewer 2 Report
The paper is interesting since the view of point of modeling. However, particularly with the results, I would like to know, how can the authors validate the data obtained? On the paper there is not a comparison with previous studies. Also, what is the main contribution of your paper? Is not clear. Results indicates that in q = 4 the total work extraction of the cycle is highest and for q = 2, there is the lowest. But, what indicates this result? Is necessary strengthen your conclusions.
Author Response

(The authors gave the same response as above.)

Round 2
Reviewer 2 Report
I don´t have comments